# Adiposity and High Blood Pressure during Childhood: A Prospective Analysis of the Role of Physical Activity Intensity and Sedentary Time in the GECKO Drenthe Cohort

**DOI:** 10.3390/ijerph17249526

**Published:** 2020-12-19

**Authors:** Rikstje Wiersma, Esther Hartman, Hendrika Marike Boezen, Eva Corpeleijn

**Affiliations:** 1Department of Epidemiology, University Medical Center Groningen, University of Groningen, P.O. Box 30.001, 9700 RB Groningen, The Netherlands; h.m.boezen@umcg.nl (H.M.B.); e.corpeleijn@umcg.nl (E.C.); 2Center for Human Movement Sciences, Section F, University Medical Center Groningen, University of Groningen, P.O. Box 196, 9700 AD Groningen, The Netherlands; e.hartman@umcg.nl

**Keywords:** adiposity, blood pressure, childhood obesity, physical exercise, sedentary time

## Abstract

Whereas in adults, physical inactivity is strongly related to obesity and hypertension, in young children the evidence is inconsistent and scarce. We examined the association between physical activity (PA) behaviours at 5–6 years of age and adiposity and blood pressure (BP) at 10–11 years in 947 children (51% boys) from the Groningen Expert Center for Kids with Obesity (GECKO) Drenthe cohort. Sedentary time (ST) and light, moderate, and vigorous PA were assessed using accelerometry (ActiGraph GT3X, wear time > 600 min/day, ≥3 days). Body mass index (BMI), waist circumference (WC), and systolic and diastolic BP were measured at 5–6 and 10–11 years of age and standardized as age- and sex-adjusted (and height-adjusted, for BP) *z*-scores. Adjusted linear and logistic regression models showed that most PA behaviours were not related to standardized BMI or WC, overweightness/obesity, abdominal overweightness/obesity, standardized systolic or diastolic BP, pulse pressure, or prehypertension at 10–11 years of age. Only if children spent more time in vigorous PA was WC slightly lower (*B* (95% CI) = −0.08 (−0.16, −0.01) SD, stdβ = −0.068) and the increase in WC over the years was less (*B* (95% CI) = −0.10 (−0.18, −0.01) SD; stdβ = −0.083). To conclude, at this very young age, PA behaviours are not a strong predictor for overweightness/obesity or hypertension later in childhood.

## 1. Introduction

Obesity is an increasing problem in society. The World Health Organization refers to a global obesity epidemic, as obesity rates have almost tripled since 1975 [1,2]. In 2016, around 40% of adults were affected as being overweight, and 13% with obesity globally. The obesity rates among children are also rising. Worldwide, about 6% of children under the age of five years were affected by overweight or obesity in 2019 [3]. In the Netherlands, 12.0% of 4- to 12-year-old children were affected by overweightness or obesity in 2019 [4]. A recent paper comparing body mass index (BMI) trajectories for ages 5–19 years across countries showed that the BMI in the Netherlands is close to the world median [5].

Many children with obesity will remain affected by obesity when they grow older, and they will develop health problems that track into adulthood [6,7]. To illustrate, children that are overweight or have obesity have an increased risk of developing metabolic syndrome or cardiovascular disease in adolescence or adulthood [8,9,10,11]. Cardiovascular disease is the main cause of death in people with overweight, and this is mostly driven by hypertension [12,13]. As the development of overweightness starts at a young age, and prevention of overweightness at a young age is more effective than treatment after its onset, early prevention is key [14]. 

In order to prevent childhood overweightness, we need to improve the lifestyle behaviours of children early in life [15]. Physical activity (PA) behaviour is seen as one of the key factors in the prevention of overweightness and obesity and in reducing cardiometabolic risk. In adults, it is well-established that increasing time spent in PA and reducing sedentary time (ST) plays an important role [16,17]. In addition, these behaviours are also relevant in the prevention of hypertension, an important component of cardiometabolic health [18]. However, the influence of PA behaviours on adiposity or blood pressure (BP) within young children remains unclear. 

A review looking into the prospective associations between total PA and adiposity or BP in children between 5 and 17 years old, with most of the included studies performed around the age of 10, showed mixed results [19]. When looking at different intensities of PA, higher intensities of PA seemed to especially favourably influence the body weight of children at a later age [19]. No prospective association was observed for BP, irrespective of the intensity of PA [19]. However, 10 out of the 12 included studies had a short follow-up (0.5 to 3 years), and very few prospective studies assessing different intensities of PA have been performed in younger children. A recent review looking into the association between objectively measured PA behaviours and adiposity in younger children (between 2 and 7 years old) included seven prospective studies [20]. Four of the included studies showed associations between higher intensities of PA and adiposity [21,22,23,24], but three other studies did not [25,26,27]. None of the included studies showed an influence of ST or lower intensities of PA on adiposity. Regarding BP, two prospective studies in children around the age of 5 showed no evident associations between PA and BP [25,28]. As prospective studies in young children are scarce and previous results are inconclusive, larger studies in young children with a longer follow-up are needed. Therefore, the aim of our study was to examine the prospective associations between PA behaviours around the age of 5–6, with adiposity and BP, as well as overweightness and prehypertension, at 10–11 years of age in the population-based Groningen Expert Center for Kids with Obesity (GECKO) Drenthe birth cohort. We studied ST and different intensities of PA to provide detailed insights into the influence of PA behaviours at a young age on future health.

## 2. Materials and Methods

### 2.1. Study Design and Participants

The GECKO Drenthe study is a population-based birth cohort with a focus on early risk factors for overweightness and obesity. Details of the GECKO Drenthe study have been described elsewhere [29]. In 2006, almost 3000 pregnant women were recruited. Monitoring of the children started from the last trimester of the pregnancy and is still ongoing. At the age of 10–11 years (August 2016 to July 2018), 2299 children were measured for follow-up. For the current study, children with valid accelerometer-derived PA data (*n* = 1135) collected at ages 5–6 (January 2011 to October 2013) were included. Among these, 1017 children had data on adiposity or BP around the age of 5–6, and 1007 children had PA and adiposity or BP data around 10–11 years old. Children who had valid data on PA and one of the health outcomes (adiposity or BP) around the age of 5–6 and the age of 10–11, were included in the analyses. Written informed consent was obtained from parents, and the study was approved by the Medical Ethics Committee of the University Medical Center Groningen in accordance to the declaration of Helsinki of 1975, as revised in 1983. The study is registered at http://www.birthcohorts.net. 

### 2.2. Physical Activity

PA was measured using ActiGraph GT3X accelerometers (ActiGraph, Pensacola, FL, United States), which are reliable and valid devices to objectively measure PA volume and intensity in young children [30,31]. The ActiGraph was placed on the child’s right hip with an elastic belt and worn during all waking hours for four consecutive days, except while bathing or swimming. Data was collected using a frequency of 30 Hz and analysed with 15 s epoch recordings. Non-wearing time was defined as periods of at least 90 min with zero counts [32]. A valid measurement was defined as having at least three days with a wear time of more than 600 min per day. Sending accelerometers by post sometimes resulted in a valid wearing day (>10 h/day). These “postage days” were identified by low light activity (≤100 min/day) and excluded from the analysis. Time spent at different PA intensities was assessed using the following cut-off points: ST (≤819 cpm), light PA (820–3907 cpm), moderate PA (3908–6111 cpm), vigorous PA (≥6112 cpm), and moderate-to-vigorous PA (MVPA, ≥3908 cpm) [33]. These cut-off points were the best fit for our sample, as they were developed in children with ages closest to our age group. Data are presented as average cpm/day for total PA, and average of min/day spent in ST and light, moderate, vigorous, or MVPA. Children were grouped into least-active and most-active groups using sex-specific quartiles of time spent in MVPA (min/day). Analyses were adjusted for season and wear time.

### 2.3. Adiposity and Blood Pressure

BMI and waist circumference (WC) were used as outcomes for adiposity. Height, weight, and WC at the age of 5–6 and 10–11 years were measured by trained preventive child healthcare nurses according to standardized protocols, as described previously [34]. BMI and WC were transformed into age- and sex-specific standardized *z*-scores, using Dutch growth analyser software (Growth Analyzer 3.5; Dutch Growth Research Foundation, Rotterdam, The Netherlands), with population data from 1997 as the reference [35]. Subsequently, the five-year change in zBMI and zWC was calculated as $\frac{“z-\mathrm{score}\;10–11\;\mathrm{years}”\;–\;“z-\mathrm{score}\;5–6\;\mathrm{years}”}{“\mathrm{age}\;10–11\;\mathrm{years}”\;–\;“\mathrm{age}\;5–6\;\mathrm{years}”}\times 5$. In this way, the change reflected the change over a period of exactly five years for each child. Children were classified as being affected with underweight, normal weight, overweight, or obese, using the age- and sex-specific cut-offs for children based on Cole et al. [36]. Abdominal overweightness and obesity was determined based on the 90th and 95th percentiles for *z*-scores. 

BP at the age of 5–6 and 10–11 years was measured by trained preventive child healthcare nurses in three consecutive measurements with a digital automatic BP monitor (Omron M3) and an appropriate cuff size. The child was sitting for 5 min before the BP was measured. For the analyses, mean systolic BP (SBP) and diastolic BP (DBP) were standardized for age, sex, and height, using LMS (lambda-mu-sigma) box Cox transformations [37,38]. Prehypertension and hypertension was determined based on the age-, sex-, and height-specific 90th and 95th percentiles, respectively, from the guidelines of the American Academy of Pediatrics (AAP; 2017) [39]. The pulse pressure (PP) was calculated as the difference between mean SBP and DBP. Data are presented as median (5th–95th percentile).

### 2.4. Other Factors

With regard to socio-economic status, maternal education level was obtained from questionnaires filled in by the parents during pregnancy. We classified the following groups: (1) no education–lower general secondary education, (2) senior secondary vocational education–higher general secondary education/pre-university education, and (3) higher vocational education–university.

### 2.5. Statistical Analysis

The statistical analyses were performed using IBM SPSS Statistics version 23 (IBM Corp., Armonk, NY, USA). Differences in adiposity or BP measures at the ages of 5–6 and 10–11 years were assessed using paired *t*-tests, Wilcoxon signed-rank tests, and χ^2^-tests. Separate regression models for minutes per day in ST, as well as light, moderate, vigorous, and MVPA were performed. As a determinant, vigorous PA needed log-transformation to obtain normal distribution of residuals in the regression analyses. Outcomes for adiposity included zBMI and zWC at 10–11 years, and change in the zBMI and zWC over the years as continuous outcomes, as well as overweightness or obesity (yes/no) and abdominal overweightness or obesity based on WC (yes/no) as dichotomous outcomes. With regard to BP, we used zSBP, zDBP, and PP as continuous outcomes, and prehypertension for SBP and DBP (>90th percentile AAP 2017) at 10–11 years as dichotomous outcomes. Furthermore, based on quartiles of MVPA at the age of 5–6, we examined differences in overweightness, abdominal overweightness, or prehypertension between least-active and most-active children. For this purpose, children were grouped into least active and most active groups using sex-specific quartiles of time spent in MVPA (min/day). All linear and logistic regression models were adjusted for sex, age at adiposity or BP measurement, age at PA measurement, season of PA measurement, accelerometer wear time, and maternal education level. In addition, we adjusted for zBMI at 5–6 years or zWC at 5–6 years, where applicable. BP models were additionally adjusted for cuff size and zSBP or zDBP at 5–6 years. Resulting *p*-values of <0.05 were considered statistically significant.

## 3. Results

### 3.1. General Characteristics

In total, 947 children had valid data on all relevant determinants and outcomes, including PA and adiposity or BP around the age of 5–6, and data on adiposity or BP at the age of 10–11. Table 1 shows the descriptive characteristics and PA behaviours of the children. Approximately 26% of the mothers had a low education level, 31% had a middle education level, and 43% had a high education level. Children spent on average more than 6 h per day sedentary and 65 min per day in MVPA at the age of 5–6 years. An overview of adiposity and BP measures of the children at both time points is shown in Table 2.

### 3.2. Early Life PA Behaviours and Future Adiposity or Blood Pressure

The prospective associations of PA behaviours with adiposity are shown in Table 3. Time spent sedentary was not related to adiposity in later childhood. Furthermore, light PA, moderate PA, and MVPA were also not related to zBMI or zWC at 10–11 years, change in zBMI or zWC between 5–6 and 10–11 years old, or (abdominal) overweightness or obesity at 10–11 years old. Only vigorous PA was related to WC, showing that young children who spent more time in vigorous PA had a lower zWC at 10–11 years and had less of an increase in zWC over the 5-year period compared to their less active peers. 

With regard to BP, no association between early life ST and BP was found, be it zSBP, zDBP, PP, or hypertension, at 10–11 years of age (Table 4). In addition, no association was found between light PA, moderate PA, vigorous PA, or MVPA and the various BP outcomes.

Next, to investigate whether a difference in health may only exist comparing extremes of PA, we studied the children in quartiles of MVPA and compared the lowest quartile (least active 25% of children, mean MVPA = 37.9 (range: 11.9–53.1) min/day) to the highest quartile (most active 25% of children, mean MVPA = 97.1 (range: 71.8–183.4) min/day). Figure 1 shows the odds ratios and 95% confidence intervals from the logistic regression models. The least active children did not have significantly higher odds of becoming affected by overweightness or obesity, or by abdominal overweightness or obesity compared to the most active children. In addition, the least active children did not have significantly higher odds of developing prehypertension than the most active children.

## 4. Discussion

In this study, we examined the prospective association of PA behaviours at 5–6 years of age with adiposity and BP at 10–11 years of age. ST or the different intensities of PA, except for vigorous PA, were not related to BMI, WC, or BP 5 years later.

Young children around 5–6 years of age who spent more time in vigorous PA, the highest intensity of PA, showed a lower standardized WC at 10–11 years and less increase in WC compared to their less active peers. Examples of vigorous PA in young children are active games involving running and chasing, fast bicycle riding, running, and sports [40,41]. However, the association between vigorous PA and WC in our study was not strong, as the effect estimates of the association were rather small. This was further confirmed when we compared the least active 25% of children to the most active 25% of children. We found that children who were relatively inactive had no higher risk of developing general overweightness or abdominal overweightness when compared to the most active children. In comparison, Metcalf et al. studied the influence of PA on children’s body weight as well. They examined MVPA around the age of five and three-year changes in BMI, skinfold thickness, or WC, and found no significant associations either [25].

Although the evidence for an association between physical (in)activity and obesity is inconsistent and scarce in young children, it seems more evident in older children. One study in children aged 9–15 years showed that more time spent sedentarily was associated with increases in BMI over a six-year period [42]. Another study in children around 10 years of age revealed that more time spent in MVPA was associated with a lower WC [43]. One way to explain this is by the changing environments when children grow older. From infancy to approximately 5 years of age, children become less dependent and less restricted in their abilities to move around freely, which results in more time spent in PA [44,45]. For infants, it has been shown that avoiding restrictions to move freely is important for healthy growth [46]. The influence of restrictions to move freely on adiposity can be observed in the years following infancy, when we compare two cross-sectional studies from different cultures. In a study in 4- to 6-year-old Dutch children, PA was not related to weight status, whereas in a comparable study among 3- to 6-year-old Chinese children, substantial differences in activity between overweight and non-overweight children were found [47,48]. Both studies used the same methods, procedures, and accelerometer cut-offs. However, children in China were much more restricted in their spontaneous PA than in the Netherlands, due to a different school system, including an obligatory nap around midday. In the current study, all children were attending kindergarten (referred to as “Group 1” or “Group 2” in the Dutch educational system) at the time of the PA measurements. In this early childhood education program, the children have ample opportunity to move around freely. Therefore, associations between PA and adiposity may become stronger when the children become more restricted to move around freely and the environment becomes more obesogenic. Another explanation for a more evident association in older children is reversed causality. There is a growing understanding that the relation between PA and adiposity might be the other way around, at least early in life. Increasing evidence suggests that childhood overweightness may be a cause of low activity levels, rather than a consequence [49,50]. Taken together, low levels of PA at a young age may not be related to health outcomes like adiposity, but we expect the associations between PA and adiposity to become stronger when children become older.

PA was hypothesized to affect adiposity, but it may also have potentially direct effects on cardiometabolic health, independent of adiposity [28]. In our study population, the prevalence of prehypertension and hypertension was rather high. A number of guidelines are available to define hypertension, each resulting in a different prevalence of hypertension [38,51]. We decided to define prehypertension and hypertension based on the 90th and 95th percentile of the guidelines from the AAP 2017, respectively [52]. Recent literature has shown that the AAP 2017 guidelines result in a higher prevalence of hypertension compared to two other frequently used guidelines [53,54]. Nevertheless, the AAP 2017 guidelines are expected to better recognize youth that are at high risk for cardiovascular disease, and have been therefore recommended in the recent literature [52,55]. In addition, the Dutch children in our study were relatively tall compared to the reference population of the AAP 2017 guidelines, and even though the hypertension cut-offs are height-dependent, the tallness of our study population may have resulted in an overestimation of hypertension. This said, no effect of ST or PA at 5–6 years on BP 5 years later was observed in our study. Accordingly, previous prospective studies in children have also found no evident associations between objectively measured PA behaviours and BP either [25,28,56,57,58]. In young children, one prospective study in five-year-old children from the United Kingdom with a three-year follow up did not show an evident association between PA and changes in BP [25]. Only the more active boys had a slightly smaller increase in mean arterial BP over the 3 years, compared to the less active boys [25]. A second prospective study in five- to seven-year-old children showed that children who engaged in more total PA had a lower DBP 2 years later compared to their peers, but no association was found for SBP, and MVPA was not related to BP [28]. In older children, one prospective study in 9- to- 15-year-old children, taking into account light PA, moderate PA, and vigorous PA, showed that in boys only, more time spent in vigorous PA was related to decreased SBP 2 years later [56]. A second prospective study found an association between PA at 12 years and DBP, but not SBP, at 14 years of age [57]. A third prospective study in 8- to 11-year-old children found no association between ST, total PA, or MVPA and mean arterial BP [58]. In cross-sectional studies, however, PA was associated with a high BP in about 50% of the studies [19]. It might be that children first become affected by obesity before they develop a high BP, as previous studies have shown that children with obesity have a higher BP compared to children who are not affected by obesity [10,11]. In addition, associations may be stronger if a composite score for cardiometabolic health is used, as different components of cardiometabolic health are likely to cluster in children who are sedentary and affected by obesity [59]. Indeed, a previous prospective study in five- to seven-year-old children showed that metabolic status (mostly insulin resistance and plasma lipids) improved over a three-year follow-up time in more active children, whereas it worsened in the less active children [25]. Taken together, low levels of PA at young age are not a strong determinant of elevated BP later in childhood.

Despite our modest findings, we do not conclude that young children’s PA behaviours are not relevant for health. Young children’s PA behaviours are important, as they are likely to track into later life [60]. Moreover, early life PA is related to improved psychosocial health, increased fitness, and improved bone and skeletal health outcomes, as well as better motor and cognitive development [61]. Therefore, even though PA behaviours at a young age may not be one of the most important drivers in the development of childhood overweightness or obesity, PA is important at young age.

There are a number of strengths in our study. Firstly, we specified our analyses for ST and different intensities of PA, providing detailed insights in the influence of PA on children’s health. Secondly, this is one of a few studies examining this relationship in young children, and may be the only study to date using a large population-based cohort with a long follow-up. Thirdly, the PA data was analysed using 15 s epoch recordings. Small epoch recordings are recommended for an accurate assessment of different PA intensities in younger children, because of the typically sporadic and intermittent PA behaviours [62,63,64]. A limitation of our study is the use of BMI and WC as outcomes for adiposity, instead of using body fat percentage. A recent review found more supporting evidence for percentage of body fat than for BMI or WC [20]. Another limitation is that children who did not participate in the activity measurements differed from the participating children in terms of maternal education level, BMI, and WC. Nevertheless, given the small differences in BMI and WC, we do not expect this to have affected our overall results. In this paper, we did not study other lifestyle factors like diet, sleep, and screen time. Since the influence of PA behaviours at a young age seem minimal, future studies should examine the influence of these lifestyle factors at a young age on children’s health.

## 5. Conclusions

To conclude, the influence of PA behaviours at a young age on adiposity or BP later in childhood are minimal, as we found no clinically relevant associations between early life ST or PA or and adiposity or BP 5 years later. Nevertheless, vigorous PA at a young age is potentially important for healthy growth and development. Moreover, it is still important to stimulate PA at a young age, because of its importance for other health outcomes and developmental domains like children’s motor and cognitive development.

## Figures and Tables

**Figure 1 ijerph-17-09526-f001:**
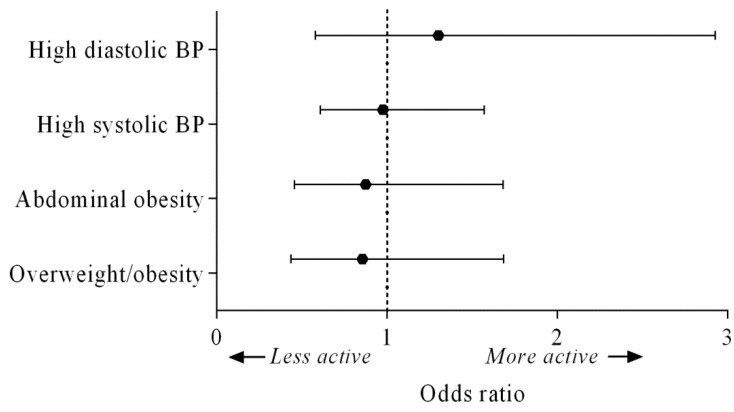
The odds of developing overweightness or obesity, abdominal overweightness or obesity, or prehypertension at the age of 10–11. The figure shows the odds ratios with 95% confidence intervals from the logistic regression models. The most active children were used as reference group. BP: blood pressure.

**Table 1 ijerph-17-09526-t001:** Descriptive characteristics and physical activity behaviours of the children.

Descriptive Characteristics	5–6 Years	10–11 Years
Sex (% boys)	483 (51.0%)	-
Age (years)	5.8 (4.8, 6.1)	10.6 (10.0, 11.4)
Height (cm)	118.6 (110.5, 137.0)	148.0 (137.0, 159.7)
Physical Activity		
Total PA (cpm)	1351.1 ± 314.4	
ST (min/day)	372.4 ± 55.5	
Light PA (min/day)	265.0 ± 38.3	
Moderate PA (min/day)	45.4 ± 15.3	
Vigorous PA (min/day)	16.7 [5.8; 42.5]	
MVPA (min/day)	64.9 ± 24.9	

Abbreviations: PA, physical activity; MVPA, moderate-to-vigorous physical activity; cpm, counts per minute; Data are shown as *n* (%), median (5th, 95th percentile), or mean ± SD.

**Table 2 ijerph-17-09526-t002:** Adiposity and blood pressure measurements of the children.

Adiposity and BP Measures	5–6 Years	10–11 Years	*p*-Value ^d^
Body mass index (*n* = 915)			
BMI (kg/m^2^)	15.7 (14.0, 18.3)	17.1 (14.5, 23.0)	<0.001
zBMI	0.16 (−1.1, 1.5)	0.12 (−1.5, 2.0)	0.182
Weight status ^a^			<0.001
Underweight	54 (5.9%)	118 (12.9%)	
Normal weight	789 (86.2%)	656 (71.7%)	
Overweight	64 (7.0%)	118 (12.9%)	
Obesity	8 (0.9%)	23 (2.5%)	
Waist circumference (*n* = 749)			
WC (cm)	54.0 (48.5, 61.0)	62.0 (55.0, 78.3)	<0.001
zWC	0.39 (−1.3, 1.8)	0.30 (−1.3, 2.2)	0.841
Abdominal overweight/obesity ^b^			<0.001
<90th percentile	644 (86.0%)	616 (82.3%)	
90th–95th percentile	64 (8.5%)	49 (6.5%)	
≥95th percentile	41 (5.5%)	84 (11.2%)	
Blood pressure (*n* = 749)			
SBP (mm Hg)	103.3 (91.0, 117.0)	108.0 (92.4, 124.8)	<0.001
DBP (mm Hg)	61.3 (50.0, 73.7)	63.3 (51.7, 76.0)	<0.001
*z*-SBP	0.56 (−0.6, 1.8)	0.23 (−1.2, 1.8)	<0.001
*z*-DBP	0.23 (−0.8, 1.4)	0.08 (−1.0, 1.2)	<0.001
PP (mm Hg)	41.7 (31.2, 54.3)	44.0 (33.0, 57.7)	<0.001
Hypertension for SBP ^c^			<0.001
Normal BP	522 (69.7%)	525 (70.1%)	
Prehypertension	78 (10.4%)	106 (14.1%)	
Hypertension	149 (19.9%)	118 (15.8%)	
Hypertension for DBP ^c^			<0.001
Normal BP	612 (81.7%)	690 (92.1%)	
Prehypertension	68 (9.1%)	34 (4.6%)	
Hypertension	69 (9.2%)	25 (3.3%)	

Abbreviations: BMI, body mass index; WC, waist circumference; BP, blood pressure; SBP, systolic blood pressure; DBP, diastolic blood pressure; PP, pulse pressure. Data are shown as *n* (%) or median (5th, 95th percentile). ^a^ Underweight, normal weight, overweight, or obesity was defined based on Cole et al. [36]. ^b^ Abdominal overweightness or obesity was classified using the 90th and 95th percentiles for *z*-scores. ^c^ Hypertension was defined using the age-, sex-, and height-specific 90th (prehypertension) and 95th (hypertension) percentiles from the American Academy of Pediatrics (AAP) guidelines. ^d^ Statistical analyses was performed using paired *t*-tests, Wilcoxon signed-rank tests, and χ^2^-tests.

**Table 3 ijerph-17-09526-t003:** Prospective associations of physical activity behaviours with adiposity.

**Body Mass Index (BMI; *n* = 915)**					
	**zBMI 10–11 Years ^d^**		**Change zBMI^e^**		**Overweightness/Obesity ^d,f^**
	**stdβ**	**B (95% CI)**	**stdβ**	**B (95% CI)**	**Odds Ratio (95% CI)**
Sedentary time ^a^					
Unadjusted model	−0.008	−0.001 (−0.01, 0.01)	−0.014	−0.002 (−0.01, 0.01)	0.999 (0.97, 1.03)
Adjusted model ^c^	−0.003	−0.001 (−0.01, 0.01)	−0.004	−0.001 (−0.01, 0.01)	1.01 (0.96, 1.06)
Light physical activity ^a^					
Unadjusted model	0.065	0.02 (0.0001, 0.04) ^h^	0.027	0.01 (−0.01, 0.02)	1.03 (0.98, 1.08)
Adjusted model ^c^	0.004	0.001 (−0.01, 0.01)	0.005	0.001 (−0.01, 0.01)	0.98 (0.92, 1.05)
Moderate physical activity ^a^					
Unadjusted model	0.057	0.04 (−0.01, 0.08)	0.039	0.02 (−0.01, 0.05)	1.07 (0.96, 1.21)
Adjusted model ^c^	0.008	0.01 (−0.03, 0.04)	0.012	0.01 (−0.03, 0.04)	1.01 (0.87, 1.17)
Vigorous physical activity ^b^					
Unadjusted model	−0.024	−0.03 (−0.10, 0.05)	0.007	0.01 (−0.05, 0.06)	0.90 (0.73, 1.10)
Adjusted model ^c^	−0.007	−0.01 (−0.06, 0.05)	−0.009	−0.01 (−0.06, 0.05)	0.92 (0.71, 1.20)
Moderate-to-vigorous physical activity ^a^					
Unadjusted model	0.020	0.01 (−0.02, 0.04)	0.025	0.01 (−0.01, 0.03)	1.01 (0.94, 1.09)
Adjusted model ^c^	−0.0002	−0.0001 (−0.02, 0.02)	0.001	0.0004 (−0.02, 0.02)	0.999 (0.91, 1.10)
**Waist Circumference (WC; *n* = 749)**					
	**zWC 10–11 years ^d^**		**Change zWC ^e^**		**Abdominal Obesity ^d,g^**
	**stdβ**	**B (95% CI)**	**stdβ**	**B (95% CI)**	**Odds Ratio (95% CI)**
Sedentary time ^a^					
Unadjusted model	−0.030	−0.01 (−0.02; 0.01)	−0.003	−0.001 (−0.01; 0.01)	0.98 (0.95; 1.01)
Adjusted model ^c^	−0.019	−0.004 (−0.02; 0.01)	−0.010	−0.002 (−0.02; 0.01)	0.98 (0.93; 1.02)
Light physical activity ^a^					
Unadjusted model	0.057	0.02 (−0.004; 0.03)	0.055	0.01 (−0.004; 0.03)	1.04 (0.99; 1.09)
Adjusted model ^c^	0.039	0.01 (−0.01; 0.03)	0.034	0.01 (−0.01; 0.03)	1.03 (0.97; 1.10)
Moderate physical activity ^a^					
Unadjusted model	0.022	0.02 (−0.03; 0.06)	0.018	0.01 (−0.04; 0.06)	1.08 (0.95; 1.22)
Adjusted model ^c^	0.005	0.003 (−0.04; 0.05)	−0.011	−0.01 (−0.06; 0.04)	1.14 (0.98; 1.34)
Vigorous physical activity ^b^					
Unadjusted model	−0.052	−0.06 (−0.15; 0.03)	−0.056	−0.06 (−0.15; 0.02)	0.85 (0.69; 1.06)
Adjusted model ^c^	−0.068	−0.08 (−0.16; −0.01) ^h^	−0.083	−0.10 (−0.18; −0.01) ^h^	0.87 (0.66; 1.15)
Moderate-to-vigorous physical activity ^a^					
Unadjusted model	−0.008	−0.003 (−0.03; 0.03)	−0.004	−0.002 (−0.03; 0.03)	0.998 (0.92; 1.08)
Adjusted model ^c^	−0.021	−0.01 (−0.04; 0.02)	−0.033	−0.01 (−0.04; 0.02)	1.02 (0.93; 1.13)

For continuous outcomes, analyses were performed using linear regression models, and standardized β and unstandardized *B* with 95% confidence interval are presented. For dichotomous outcomes, logistic regression models were used, and odds ratios with 95% confidence interval are presented. ^a^ Physical activity and sedentary time are expressed as 10 min/day. ^b^ Vigorous physical activity was log-transformed. ^c^ All analyses were adjusted for the age of physical activity measurement, sex, maternal education level, season of physical activity measurement, and wear time. ^d^ For these outcomes, the analyses were additionally adjusted for age of outcome measurement and baseline *z*-score. ^e^ The change in zBMI and zWC represents the change between 5–6 and 10–11 years. ^f^ At 10–11 years, 141 children were affected by overweightness or obesity, based on Cole et al. [36]. ^g^ At 10–11 years, 133 children were affected by abdominal overweightness or obesity, defined using the 90th percentile for *z*-scores; ^h^
*p* < 0.05.

**Table 4 ijerph-17-09526-t004:** Prospective associations of physical activity behaviours with blood pressure.

	z-Systolic BP 10–11 Years		z-Diastolic BP 10–11 Years		Pulse Pressure 10–11 Years		High SBP ^d^	High DBP ^d^
	stdβ	B (95% CI)	stdβ	B (95% CI)	stdβ	B (95% CI)	Odds Ratio (95% CI)	Odds Ratio (95% CI)
Sedentary time ^a^								
Unadjusted model	0.005	0.001 (−0.01, 0.01)	0.018	0.002 (−0.01, 0.01)	−0.012	−0.02 (−0.11, 0.08)	0.995 (0.97, 1.02)	1.02 (0.97, 1.07)
Adjusted model ^c^	−0.037	−0.01 (−0.02, 0.01)	−0.048	−0.01 (−0.02, 0.004)	0.005	0.01 (−0.10, 0.11)	0.99 (0.95, 1.02)	0.997 (0.94, 1.06)
Light physical activity ^a^								
Unadjusted model	0.032	0.01 (−0.01, 0.03)	0.016	0.003 (−0.01, 0.02)	0.029	0.06 (−0.08, 0.19)	1.02 (0.98, 1.06)	0.97 (0.90, 1.04)
Adjusted model ^c^	0.031	0.01 (−0.01, 0.02)	0.045	0.01 (−0.01, 0.02)	−0.008	−0.02 (−0.16, 0.13)	1.01 (0.97, 1.06)	1.01 (0.93, 1.09)
Moderate physical activity ^a^								
Unadjusted model	0.016	0.01 (−0.03, 0.05)	−0.013	−0.01 (−0.04, 0.03)	0.033	0.16 (−0.19, 0.51)	1.01 (0.91, 1.12)	0.97 (0.81, 1.16)
Adjusted model ^c^	0.035	0.02 (−0.02, 0.06)	0.045	0.02 (−0.01, 0.05)	−0.003	−0.01 (−0.36, 0.33)	1.04 (0.92, 1.17)	1.06 (0.86, 1.30)
Vigorous physical activity ^b^								
Unadjusted model	0.006	0.01 (−0.07, 0.08)	−0.029	−0.02 (−0.08, 0.03)	0.038	0.33 (−0.29, 0.94)	1.00 (0.83, 1.20)	0.90 (0.66, 1.22)
Adjusted model ^c^	0.006	0.01 (−0.07, 0.08)	−0.011	−0.01 (−0.06, 0.05)	0.017	0.14 (−0.45, 0.73)	1.02 (0.83, 1.26)	0.87 (0.62, 1.21)
Moderate-to-vigorous physical activity ^a^								
Unadjusted model	0.011	0.004 (−0.02, 0.03)	−0.016	−0.004 (−0.02, 0.02)	0.030	0.09 (−0.13, 0.31)	1.01 (0.94, 1.07)	0.97 (0.87, 1.08)
Adjusted model ^c^	0.025	0.01 (−0.02, 0.04)	0.025	0.01 (−0.01, 0.03)	0.003	0.01 (−0.21, 0.22)	1.02 (0.95, 1.10)	0.99 (0.87, 1.12)

Abbreviations: BP, blood pressure; SBP, systolic blood pressure; DBP, diastolic blood pressure. For continuous outcomes, analyses were performed using linear regression models, and standardized β and unstandardized *B* with 95% confidence interval are presented. For dichotomous outcomes, logistic regression models were used, and odds ratios with 95% confidence interval are presented. A total of 749 children were included in the analyses. ^a^ Physical activity and sedentary time expressed as 10 min/day. ^b^ Vigorous physical activity was log-transformed. ^c^ All analyses were adjusted for age of physical activity measurement, outcome measurement, sex, baseline measurement, cuff size, maternal education level, season of physical activity measurement, and wear time. ^d^ Prehypertension at 10–11 years was defined using the age-, sex-, and height-specific 90th percentile from the American Academy of Pediatrics (AAP) guidelines. In total, 224 children and 59 children were classified as having high systolic and diastolic blood pressure, respectively.

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
