# Peer review of "Adiposity and High Blood Pressure during Childhood: A Prospective Analysis of the Role of Physical Activity Intensity and Sedentary Time in the GECKO Drenthe Cohort"

_ijerph, 2020, doi:10.3390/ijerph17249526_

Round 1
Reviewer 1 Report
Thank you for providing me the opportunity to review this manuscript. As a registered dietitian with research speciality for lifestyle habits (nutrition, physical activity, sedentary time, screen time, sleep) among children, I read this manuscript with interest. The authors stated that sedentary time or the different intensities of physical activity at 5-6 years were not related to body mass index, waist circumference, or blood pressure 5 years later, except for vigorous physical activity. The findings suggest that, at this very young age, physical activity behaviours are not a strong predictor for overweight/obesity or hypertension later in childhood. The points below might strengthen the manuscript.
Introduction :
Lifestyle habits other than physical activity and sedentary time, like nutrition and sleep, also represent key factors in the prevention of overweight and obesity and in reducing cardiometabolic risk.
Methods :
Could you discuss a little bit more about the choice of the cut-off points for physical activity, other than ''were the best fit for our age group''? According to this paper [Migueles JH et al. Accelerometer Data Collection and Processing Criteria to Assess Physical Activity and Other Outcomes: A Systematic Review and Practical Considerations. Sports Med. 2017;47(9):1821-1845], cut-off points by Jimmy et al. would be more appropriate [Jimmy G et al. Development and validation of GT3X accelerometer cut-off points in 5- to 9-year-old children based on indirect calorimetry measurements. Schweizerische
Zeitschrift fur Sport und Sport. 2013;61:37–43].
Do you have national guidelines in The Netherlands for physical activity and sedentary time among children? It would be more appropriate to use this recommendation comparatively to sex-specific quartiles.
Could you give some examples of activities from different intensities among this specific age group? For example, in Canada, ''energetic play'' is used to refer to moderate- to vigorous-intensity physical activity (MVPA). It is more
appropriately contextualized for the early years and refers to activities for young children that get them working hard, breathing heavily and feeling warm: ball games, activities in the park (riding a tricycle or bike), water activities, tag... [https://www.csep.ca/CMFiles/Guidelines/24hrGlines/24HourGuidelinesGlossary_2017.pdf].
Results :
Presentation of the tables (Table 1, Table 2, Table 3) should be improved, for overall clarity. There is a lot of information presented.
Discussion :
I really appreciated this part of the discussion : ''Despite our modest findings, we do not conclude that young children’s PA behaviours are not relevant for health. Young children’s PA behaviours are important, as they are likely to track into later life. Moreover, early life PA is related to improved psychosocial health, increased fitness and improved bone- and skeletal health outcomes, as well as better motor and cognitive development. Therefore, even though PA behaviours at a young age may not be one of the most important drivers in the development of childhood overweight or obesity, PA is important at young age.''
A better ''strengths and limitations'' section is needed. Physical activity data analysed with 15 second epoch recordings is a strength (short epochs enable the devices to capture small bouts of vigorous PA, which is typical for
this age group, while obtaining high accuracy in their validation). Moreover, the lack of information about diet and other limitations should be described fully.
Author Response
We thank the reviewer for his/her overall appreciation of our paper. We have clarified the points brought up by the reviewer below:
Comments and Suggestions for Authors
Thank you for providing me the opportunity to review this manuscript. As a registered dietitian with research speciality for lifestyle habits (nutrition, physical activity, sedentary time, screen time, sleep) among children, I read this manuscript with interest. The authors stated that sedentary time or the different intensities of physical activity at 5-6 years were not related to body mass index, waist circumference, or blood pressure 5 years later, except for vigorous physical activity. The findings suggest that, at this very young age, physical activity behaviours are not a strong predictor for overweight/obesity or hypertension later in childhood. The points below might strengthen the manuscript.
Introduction :
Lifestyle habits other than physical activity and sedentary time, like nutrition and sleep, also represent key factors in the prevention of overweight and obesity and in reducing cardiometabolic risk.
Indeed, physical activity and sedentary time are not the only key factors in the prevention of overweight and obesity. Within the GECKO Drenthe cohort, we have data on diet and sleep as well. However, these lifestyle habits are known to influence each other and therefore more sophisticated analyses would be required. For now, we were primarily interested in detailed analyses of physical activity and sedentary time. The influence of diet and sleep, and the clustering of the lifestyle habits, is beyond the scope of this paper. It is definitely worth investigating this in another study.
Methods :
Could you discuss a little bit more about the choice of the cut-off points for physical activity, other than ''were the best fit for our age group''? According to this paper [Migueles JH et al. Accelerometer Data Collection and Processing Criteria to Assess Physical Activity and Other Outcomes: A Systematic Review and Practical Considerations. Sports Med. 2017;47(9):1821-1845], cut-off points by Jimmy et al. would be more appropriate [Jimmy G et al. Development and validation of GT3X accelerometer cut-off points in 5- to 9-year-old children based on indirect calorimetry measurements. Schweizerische Zeitschrift fur Sport und Sport. 2013;61:37–43].
Many accelerometer cut-points are available. In a recent review among objectively measured physical activity in children between 2 and 7 years old, we observed 37 different accelerometer cut-points used[1]. This shows that many different cut-points are used in this age group. The cut-points by Butte et al. (2014) were frequently used in triaxial accelerometers.
The cut-points by Butte et al. (2014) were the best fit four our age group, because they are developed in children with ages close to our age group. In the paper by Jimmy et al. (2013) they included 5- to 9-year old children with a mean age of 7.6 ± 1.4 years. In the paper by Butte et al. (2014) they included children aged 4.6 ± 0.9 years. The mean age of the children in our study sample is 5.7 ± 0.8 years. We considered that cut-points from younger children would be a better fit, as children in our study were attending kindergarten at the time of the PA measurements.
Nevertheless, irrespective of which cut-points were used, we think the results in our study would be similar. A study by Banda et al. (2016) showed that the use of different accelerometer cut points could cause a difference of 200 min estimated sedentary time or time engaged in light physical activity and approximately 50 min on moderate-to-vigorous physical activity[2]. Although exact time spent sedentary or physically active would differ when different cut-points were used, the relative differences between children would be similar. Therefore, we assume that the associations between physical activity behaviours and adiposity or hypertension would not differ if different cut-points were used.
The following was changed in the method section (page 3, lines 103-104)
'' These cut-off points were the best fit for our sample, as they were developed in children with ages closest to our age group.”.
[1]. Wiersma R, Haverkamp BF., van Beek JH, et al. Unravelling the association between accelerometer‐derived physical activity and adiposity among preschool children: A systematic review and meta‐analyses. Obesity reviews. 2020;21(2):e12936. https://doi.org/10.1111/obr.12936
[2]. Banda JA, Haydel KF, Davila T, et al. Effects of varying epoch lengths,wear time algorithms, and activity cut‐points on estimates of child sedentary behavior and physical activity from accelerometer data. PLoS ONE. 2016;11(3):e0150534. https://doi.org/10.1371/journal.pone.0150534
Do you have national guidelines in The Netherlands for physical activity and sedentary time among children? It would be more appropriate to use this recommendation comparatively to sex-specific quartiles.
Dutch guidelines for physical activity and sedentary time are available. The guidelines are provided by the Dutch Health Council. For children between 4 and 18 years old, it is recommended to engage in moderate-to-vigorous physical activity for at least 1 hour per day. In addition, children should minimize the time spent sitting.
We did briefly consider using these Dutch guidelines for physical activity to compare the active and less-active children, but found them to be not appropriate given our research question. We preferred to maximise the contrast between the groups by looking at the extremes of physical activity. Therefore we choose to use quartiles based on moderate-to-vigorous physical activity (MVPA)instead of using the physical activity guidelines. To add, if the Dutch guidelines were used, the results would be similar. Children who did not spent at least 1 hour per day in MVPA had a mean MVPA of 44.7 (range: 11.9-59.9) min/day and children with at least 1 hour per day in MVPA had a mean MVPA of 83.2 (range: 60.0-183.4) min/day. In comparison, children in the lowest quartile had a mean MVPA of 37.9 (range: 11.9-53.1) min/day and children in the highest quartile a mean MVPA of 97.1 (range: 71.8-183.4) min/day.
Could you give some examples of activities from different intensities among this specific age group? For example, in Canada, ''energetic play'' is used to refer to moderate- to vigorous-intensity physical activity (MVPA). It is more appropriately contextualized for the early years and refers to activities for young children that get them working hard, breathing heavily and feeling warm: ball games, activities in the park (riding a tricycle or bike), water activities, tag... [https://www.csep.ca/CMFiles/Guidelines/24hrGlines/24HourGuidelinesGlossary_2017.pdf].
Thank you for this excellent suggestion. We now have added examples of activities among the age group to increase the comprehensibility of our study. Since we found that only more time spent in vigorous physical activity could reduce waist circumference in children, we added examples of vigorous physical activity. We could not find examples for Dutch children specifically, therefore chose to use the examples from the Canadian 24 hour movement guidelines. The following was added to the discussion section (page 12, lines 208-209):
“Examples of vigorous PA in young children are active games involving running and chasing, fast bicycle riding, running and sports[40,41].”
Results :
Presentation of the tables (Table 1, Table 2, Table 3) should be improved, for overall clarity. There is a lot of information presented.
We understand the suggestion to improve the tables. We divided Table 1 into two tables to increase the clarity; Table 1 shows the descriptive characteristics and PA behaviours of the children and Table 2 contains the overview of adiposity and BP measures of the children at both time points. We would rather not change the content of the other tables. Although it is a lot of information, all information that belongs together in terms of content is now together (i.e. a separate table for adiposity and a separate table for blood pressure). To increase the clarity of these tables, we checked the completeness of the footnotes and added information about the statistical outcomes. The following was added to the footnotes of Table 2 and 3:
“For continuous outcomes, analyses were performed using linear regression models and standardized β and unstandardized B with 95% confidence interval were presented. For dichotomous outcomes, logistic regression models were used and odds ratio’s with 95% confidence interval were presented.”
Discussion :
I really appreciated this part of the discussion : ''Despite our modest findings, we do not conclude that young children’s PA behaviours are not relevant for health. Young children’s PA behaviours are important, as they are likely to track into later life. Moreover, early life PA is related to improved psychosocial health, increased fitness and improved bone- and skeletal health outcomes, as well as better motor and cognitive development. Therefore, even though PA behaviours at a young age may not be one of the most important drivers in the development of childhood overweight or obesity, PA is important at young age.''
A better ''strengths and limitations'' section is needed. Physical activity data analysed with 15 second epoch recordings is a strength (short epochs enable the devices to capture small bouts of vigorous PA, which is typical for this age group, while obtaining high accuracy in their validation). Moreover, the lack of information about diet and other limitations should be described fully.
Thank you for the suggestion to add the 15 second epoch recordings as a strength of the study. We added this to the strengths and limitations section. We do not consider the lack of information about diet and other lifestyle factors as a limitation of our study. As the lifestyle factors are known to influence each other, more sophisticated analyses would be required to take into account the other lifestyle factors as well. In that case, we would not be able to perform detailed analyses of physical activity and sedentary time. For now, we were primarily interested in the detailed analyses of physical activity and sedentary time, and the influence of other lifestyle factors is beyond the scope of this paper. We did add a recommendation for future research to examine the influence of diet, sleep and screen time at a young age.
The following sentences were added to the discussion section (page 13, lines 287-298):
“Thirdly, the PA data was analysed using 15 second epoch recordings. Small epoch recordings are recommended for an accurate assessment of different PA intensities in younger children, because of the typically sporadic and intermittent PA behaviours of the children[63-65]. […………].In this paper, we did not study other lifestyle factors like diet, sleep and screen time. Since the influence of PA behaviours at a young age seem minimal, future studies should examine the influence of these lifestyle factors at a young age on children’s health.
Reviewer 2 Report
This paper has shown in a simple and precise way how PA in young age can impact the risk of obesity and hypertension in later age. High number of included participants and strong methodology are important assets of this manuscript. Even though the authors didn't find that PA has influence on adiposity and blood pressure in later age, these results are of high importance.
There are only minor text revisions required:
Section discussion:
line 204 Please delete double "when we" in the text
line 217 "... might be the other way other around..."
Author Response
We thank the reviewer for his/her overall appreciation of our paper and for the text revisions. We changed the document according to the comments.
Comments and Suggestions for Authors
This paper has shown in a simple and precise way how PA in young age can impact the risk of obesity and hypertension in later age. High number of included participants and strong methodology are important assets of this manuscript. Even though the authors didn't find that PA has influence on adiposity and blood pressure in later age, these results are of high importance.
There are only minor text revisions required:
Section discussion:
line 204 Please delete double "when we" in the text
line 217 "... might be the other way other around..."
Reviewer 3 Report
Congratulations to the authors, the study is very interesting and has many strong points. The conclusions are very well based on the results and the discussion. The design is very interesting and laborious.
The only question to comment on is why the following sentence in table 1 is not put in the statistical analysis section? "d Statistical analyses was performed using paired t-tests, Wilcoxon signed-rank tests, and χ2-tests".
Author Response
Comments and Suggestions for Authors
Congratulations to the authors, the study is very interesting and has many strong points. The conclusions are very well based on the results and the discussion. The design is very interesting and laborious.
The only question to comment on is why the following sentence in table 1 is not put in the statistical analysis section? "d Statistical analyses was performed using paired t-tests, Wilcoxon signed-rank tests, and χ2-tests".
We thank the reviewer for his/her overall appreciation of our paper and for the sharp-eyed comment. We simply forgot to add the sentence to the statistical analysis section. The following was added to the statistical analysis section (page 3, lines 138-140)
“Differences in adiposity or BP measures at the age of 5-6 and at the age of 10-11 years were assessed using paired t-tests, Wilcoxon signed-rank tests and χ2-tests.”
Reviewer 4 Report
Thank you for the opportunity to review this article. The work is very interesting, but some aspects should be taken into account before publication.
Comments and suggestions for Authors:
- Keywords: It needs to be sorted in alphabetical order.
- Please correct the references according the guidelines for authors, e.g. number 42.
Introduction:
- Authors should include a literature review in the introduction. The type “article” requires to clearly provide rationale for the study (with research questions and hypotheses) and clearly identify what this study adds to the current literature on this topic. Please expand this section.
Materials and Methods:
- Why authors include exactly 947 children? In 2.1 subsection authors wrote about
- Please add inclusion and exclusion criteria.
Results:
- Table 1 – please correct; it is hard to understand for the readers
Author Response
We thank the reviewer for his/her overall appreciation of our paper. We have clarified the points brought up by the reviewer below:
Comments and Suggestions for Authors
Thank you for the opportunity to review this article. The work is very interesting, but some aspects should be taken into account before publication.
Comments and suggestions for Authors:
Keywords: It needs to be sorted in alphabetical order.
Thank you, we changed to order of the keywords.
Please correct the references according the guidelines for authors, e.g. number 42.
All references were checked and, if needed, corrected. Reference 42 and 43 (now 51 and 52) were corrected.
Introduction:
Authors should include a literature review in the introduction. The type “article” requires to clearly provide rationale for the study (with research questions and hypotheses) and clearly identify what this study adds to the current literature on this topic. Please expand this section.
We did expand the introduction by providing more information about previous prospective studies in younger children. The following was added to the introduction (page 2, lines 59-67):
“A recent review looking into the association between objectively measured PA behaviours and adiposity in younger children (between 2 and 7 years old) included seven prospective studies[20]. Four of the included studies showed associations between higher intensities of PA and adiposity[21–24], but three other studies did not[25–27]. None of the included studies showed an influence of ST or lower intensities of PA on adiposity. With regard to BP, two prospective studies in children around the age of 5 showed no evident associations between PA and BP[25,28]. As prospective studies in young children are scarce and previous results are inconclusive, larger studies in young children with a longer follow-up are needed.”
Materials and Methods:
Why authors include exactly 947 children? In 2.1 subsection authors wrote about
Please add inclusion and exclusion criteria.
We clarified the in- and exclusion criteria by adding the following sentence to subsection 2.1 (page 2, lines 83-85):
“Children who had valid data on PA and one of the health outcomes (adiposity or BP) around the age of 5-6 and the age of 10-11, were included in the analyses.”
Results:
Table 1 – please correct; it is hard to understand for the readers
We divided Table 1 into two tables to increase the clarity; Table 1 shows the descriptive characteristics and PA behaviours of the children and Table 2 contains the overview of adiposity and BP measures of the children at both time points.